# NanoBubble-Mediated Oxygenation: Elucidating the Underlying Molecular Mechanisms in Hypoxia and Mitochondrial-Related Pathologies

**DOI:** 10.3390/nano13233060

**Published:** 2023-11-30

**Authors:** Sergio M. Viafara Garcia, Muhammad Saad Khan, Ziyad S. Haidar, Juan Pablo Acevedo Cox

**Affiliations:** 1Programa de Doctorado en BioMedicina, School of Medicine, Universidad de los Andes, Santiago 8150513, Chile; 2Center of Interventional Medicine for Precision and Advanced Cellular Therapy (IMPACT), Santiago 8150513, Chile; 3Centro de Investigación e Innovación Biomédica (CiiB), Universidad de los Andes, Santiago 8150513, Chile; 4Cell for Cells, Consorcio Regenero, and TE and BioFab LAB, Universidad de los Andes, Santiago 8150513, Chile; 5School of Dentistry, Universidad el Bosque, Bogota 110121, Colombia; 6Department of Physics, Toronto Metropolitan University, Toronto, ON M5B 2K3, Canada; 7Programa de Doctorado en Ciencias Odontológicas, School of Dentistry, Universidad de los Andes, Santiago 8150513, Chile; 8BioMAT’X R&D&I (HAiDAR I+D+i) LAB, Faculty of Dentistry, Universidad de los Andes, Santiago 8150513, Chile

**Keywords:** oxygen, nanobubbles, hypoxia, mitochondria, metabolism, molecular, drug delivery, cancer, stress, innovation

## Abstract

Worldwide, hypoxia-related conditions, including cancer, COVID-19, and neuro-degenerative diseases, often lead to multi-organ failure and significant mortality. Oxygen, crucial for cellular function, becomes scarce as levels drop below 10 mmHg (<2% O_2_), triggering mitochondrial dysregulation and activating hypoxia-induced factors (HiFs). Herein, oxygen nanobubbles (OnB), an *emerging* versatile oxygen delivery platform, offer a novel approach to address hypoxia-related pathologies. This review explores OnB oxygen delivery strategies and systems, including diffusion, ultrasound, photodynamic, and pH-responsive nanobubbles. It delves into the nanoscale mechanisms of OnB, elucidating their role in mitochondrial metabolism (TFAM, PGC1alpha), hypoxic responses (HiF-1alpha), and their interplay in chronic pathologies including cancer and neurodegenerative disorders, amongst others. By understanding these dynamics and underlying mechanisms, this article aims to contribute to our accruing knowledge of OnB and the developing potential in ameliorating hypoxia- and metabolic stress-related conditions and fostering innovative therapies.

## 1. Introduction

Hypoxia is a medical condition characterized by an inadequate/insufficient oxygen supply to tissues and organs, leading to potential harm or dysfunction. It results from various pathological phenomena, including diminished cardiac output, reduced hemoglobin concentration, and an atypical vasculature network. Currently, there is a widely acknowledged consensus that restricted oxygen availability jeopardizes not only cell viability but also cellular phenotype and functionality [1]. Furthermore, inadequate oxygen supply (*hypoxia*: different levels of oxygen deprivation, from mild to severe; *hypoxic stress*: in cellular or molecular biology research, this specifically refers to a situation where cells or tissues are exposed to lower-than-optimal oxygen levels, causing them to adapt or respond to this oxygen deficiency, where they may activate specific molecular pathways, like the hypoxia-inducible factor or HiF pathway to adapt to the low oxygen environment) may disturb numerous biological processes, particularly those requiring high energy demands, such as tissue repair, restoration, and regeneration. Upon establishing hypoxia, it could lead to a grave metabolic crisis, especially in individuals with metabolic syndromes, cardiovascular diseases, acute traumas, chronic diseases, and cancer, to list a few [2,3,4,5].

Today, it is widely recognized that exposing human blood to free oxygen gas bubbles causes hemolysis [6]. Nonetheless, various approaches have been employed to increase oxygen levels and overcome hypoxia, including the utilization of oxygen microparticles, hydrogen peroxide solutions, hyperbaric oxygenation, and inhalation of gas mixtures such as carbogen (composed of 5% CO_2_ and 95% O_2_) [7]. However, they are considered ineffective due to clinical and logistical limitations. These approaches include limited gas solubility, cytotoxicity, instability, low gas pay-load, ill-defined gas-release profiles, and low cellular uptake, rendering them unsuitable for gas delivery. Consequently, an imminent need for novel methodologies and strategies arises that can offer and/or provide oxygen to tissues and organs in a manner that is both safe/biocompatible as well as cost-effective whilst being *highly* biodegradable and easily deployable or user-friendly too [7]. 

Herein, *nanobubbles*, also known as ultra-fine bubbles or sub-micron bubbles, are extremely small gas-filled spherical bodies enclosed by an interface (gas–liquid or gas–solid) with a diameter of less than 1000 nm (i.e., ranging in size from tens to hundreds of nanometers in diameter) [8,9]. In other words, nanobubbles are composed of gas, such as oxygen or nitrogen, encapsulated within a liquid, typically water, and stabilized by a thin layer of molecules at the gas–liquid interface. Nanobubbles have gained significant attention in various fields, including medicine, materials science, and environmental science due to their unique properties and potential applications. Indeed, several remarkable properties of these nanobubbles have been widely studied, such as their long-term stability and longevity [10,11,12], enhanced dissolution of gases, and increased gas interfacial diffusion [13,14]. In addition, nanobubbles can be biocompatible and have demonstrated interesting potential as suitable candidates for drug delivery purposes (drugs, genes, or other therapeutic agents, with improved release and delivery to specific tissues or cells) [7,15]. Given that an optimal oxygen-release material must demonstrate compatibility with biological systems, strategies for oxygen delivery, such as nanobubbles, have garnered substantial attention as robust mechanisms to alleviate the hypoxic stress response(s) [16,17], which is an exciting and growing area of research, development, and innovation in various scientific, medical, and industrial fields, in laboratories World-wide. 

Notably, during the past decade, there has been a remarkable increase in scientific (and technological) interest and a surge in publications in this emerging field (Figure 1), particularly concerning *oxygen nanobubbles* (OnB) and *OnB-mediated oxygenation*. OnB are considered a supplementary or adjuvant approach for drug (and gene) delivery in various conditions marked or induced by hypoxia and mitochondrial dysfunction. However, to the best of our knowledge, the existing literature seems to continue to lack a fair representation and utter/comprehensive understanding of the underlying governing mechanisms.

Nonetheless, it is widely accepted that metabolism and oxygen levels act as intricately interlinked factors that determine the physiological behavior of cells, the onset of pathologies, and the progression of chronic diseases. Among the advantages of developing oxygenation systems, selective metabolic reprogramming can target (1) tumor suppression in cancer by disrupting their ability to adapt to low oxygen conditions [18], (2) enhanced wound healing by stimulating angiogenesis and oxygen utilization in affected tissues [19], and (3) restoration of mitochondrial function in neurodegenerative diseases through promoting the alternative energy production pathways and reducing oxidative stress, i.e., most likely slowing disease progression and enhancing neuronal survival [20].

Since the first report describing nanobubbles, they have drawn much more attention regarding the development of applications when compared to the study of their fundamental mechanisms. Therefore, the existing biomedical literature dedicated to the underlying governing mechanisms remains lacking. Consequently, this review article opted to explore, in-depth, the therapeutic potential of oxygen itself, oxygenation, and the myriad of diverse possibilities offered by oxygen-infused nanobubbles, deemed critical for designing, developing, characterizing, testing, fine-tuning/optimizing, and eventually translating comprehensive, safe, and effective OnB-mediated oxygenation strategies, from bench to clinic. Herein, we emphasize that oxygen can serve a dual role: firstly, in its gaseous form, it can act as a therapeutic agent, promoting the restoration of oxygen dynamics, metabolism, and inflammatory balance; secondly, an excess of oxygen availability, locally, can disrupt the redox balance, which is an intriguing aspect harnessed by OnB to counter the hypoxic microenvironment within carcinogenic tumors (i.e., tumor microenvironment or TME). Indeed, this intervention reduces HiF-1 alpha expression, thereby enhancing treatment effectiveness, and micro-/nano-sized bubbles are employed as supplementary treatments to mitigate tumor cell resistance and metastasis. Henceforth, we conducted a thorough examination of the impact of OnB systems and OnB-mediated oxygenation strategies on the pathogenesis and prognosis of various conditions, with a particular focus on cancer and neuro-degenerative disorders, among other relevant ailments, diseases, conditions, and pathologies [5,7,21,22,23]. We critically examined the potential applications of OnB in the field of biomedicine by analyzing pivotal research articles. Essentially, our analytical literature study aimed to provide an interdisciplinary overview of the fundamental principles of nanobubbles. We then explored their utilization for oxygen delivery and highlighted broader applications in the biomedical context. Additionally, we delved into investigating the cellular and molecular impact of OnB on mitochondrial functionality and the signaling pathways associated with hypoxic cells, possibly contributing to the comprehension of the underlying biological effects of OnB systems across various eukaryotic cell types, including neurons, glial cells, tumorigenic, and cardio-respiratory cells, to list a few. Finally, a special focus is dedicated in this review article to elucidating the cross-talk between the mitochondrial metabolism and hypoxic cell response(s), which serve as the principal oxygen sensing mechanism(s) in eukaryotic (possessing a clearly defined and membrane-bound nucleus and defined chromosomes) cells, a type of highly diverse, specialized, and multifunctional cells in multi-cellular organisms.

## 2. Hypoxia and Mitochondrial Dysfunction: *Interplay and Implications in Pathology(ies)*

The interplay between hypoxia (oxygen deficiency) and mitochondrial dysfunction (perturbing the cellular energy-producing organelles) is pivotal in the development and progression of various human diseases. These intertwined processes, once considered distinct, have emerged as key players in conditions spanning cancer, cardiovascular diseases, neuro-degenerative disorders, and more. Hypoxia disrupts the balance of oxygen supply to tissues, triggering a cascade of cellular responses. Simultaneously, mitochondrial dysfunction, characterized by impaired energy production and oxidative stress, contributes to cellular damage and dysfunction. The dynamic interaction between these two phenomena amplifies disease mechanisms and influences treatment outcomes. Understanding the molecular intricacies of hypoxia–mitochondrial dysfunction crosstalk offers insights into disease pathogenesis and potential therapeutic targets. Exploring this complex interplay offers a promising avenue for advancing our understanding of disease mechanisms and developing novel interventions. Indeed, researchers and scientists are designing, developing, and evaluating innovative diagnostic and treatment strategies that target this nexus, providing hope for improved outcomes in a broad spectrum of human pathologies. 

The normal partial pressure of oxygen (PaO_2_) in arterial blood typically falls within the range of 75 to 100 mm of mercury (mmHg) when measured at sea level and standard atmospheric pressure, exhibiting variability across various tissues. Normal *tissue* oxygen levels can range from 0 to around 60 mmHg, but these values can fluctuate significantly depending on the specific tissue type and its physiological conditions. Measuring PaO_2_ directly within cells can be challenging and typically requires specialized techniques, such as microelectrode oxygen sensors or imaging methods. These measurements are valuable for understanding tissue oxygenation in various physiological and pathological conditions and can guide treatment strategies in clinical settings. In most cases, the physiological range spans from 14 to 65 mmHg, corresponding to atmospheric oxygen levels of 2–9% [24]. Distinct PaO_2_ gradients are also evident across both superficial and deep cellular layers, displaying diverse characteristics from vascular lumen to perivascular tissues (Figure 2A) [24,25]. On the other hand, cellular hypoxia occurs when the oxygen level is in the range of 10–16 mmHg (1–2%), and it depends on the type of tissue [26]. Once the demand for oxygen exceeds its supply, the oxygen level decreases (herein known as hypoxia) in local tissues and/or peripherical tissues, leading to a metabolic crisis [27,28] (Figure 2B,C). 

Herein, the hypoxia-inducible factor or HiF is a heterodimeric transcription factor that plays a central role in cellular adaptation to low oxygen levels. It is considered a master regulator of the cellular response to hypoxia and plays a crucial role in maintaining oxygen homeostasis in various tissues and cells [29]. Briefly, the HiF is composed of two sub-units: HiF-α (alpha) and HiF-β (beta). The alpha subunit is oxygen-sensitive and exists in different isoforms, with HiF-1α and HiF-2α being the most well-known. In normoxic (oxygen-rich) conditions, prolyl hydroxylase enzymes modify HiF-α, marking it for degradation. However, in hypoxic conditions, this degradation is inhibited, allowing HiF-α to accumulate and form a heterodimer with HiF-β. This active HiF complex then translocates to the cell nucleus, where it binds to specific DNA sequences known as hypoxia-response elements or HREs and initiates the transcription of genes involved in various adaptive responses to low oxygen, including angiogenesis, erythropoiesis, glycolysis, and more. It is noteworthy to mention that the HiF ability to regulate the expression of genes involved in oxygen homeostasis and adaptation to hypoxia makes it a critical factor in various physiological and pathological processes, including embryogenesis, ischemic diseases, cancer, and metabolic disorders. Understanding the role of HiF and its downstream targets is essential for gaining insights into how cells and organisms respond to changing oxygen levels and for developing therapeutic strategies to address related diseases and conditions [29]. Remember that once the oxygen decreases below physiological levels, cells trigger a hypoxic cell response, characterized by the HiF-1α stabilization and its nuclear transcription (Figure 3A). Consequently, the translocation of HiF-1α leads to the activation of approximately 400 target genes. This activation influences metabolic processes by upregulating proteins associated with glycolysis while concurrently suppressing oxygen-dependent pathways, particularly those related to mitochondrial metabolism [30,31]. These findings establish a direct connection between mitochondria and HiF pathway, highlighting interdependence in response to changes in oxygen supply and demand. 

Hypoxia exerts diverse effects on mitochondria, but mitochondria also play a role in modulating the cellular response to hypoxia [27,32]. Mitochondria serve as the primary oxygen sensors within cells, and their function is profoundly impacted by decreased oxygen levels. Concurrently, hypoxia influences various mitochondrial processes, including fusion, fission, mitophagy, and oxidative phosphorylation (OXPHOS) [27]. Intriguingly, the inhibition of mitochondrial function can reverse the activation of the HiF pathway induced by hypoxia [32]. Such observations prompted researchers to propose a role for the mitochondria in metabolic re-programing and in modulating HiF activation. The use of liposomes loaded with an anti-tumor drug to inhibit mitochondrial complexes I and II showed a significant reduction in OXPHOS metabolism and oxygen consumption [18]. Consequently, higher oxygen levels inside the cell are critical to reverse HiF activation. In general, these strategies have been reported to sensitize radiotherapy or photodynamic therapy (PDT) in tumors via enhancing selective DNA damage [18,33] rather than endogenous O_2_ delivery (perfluorocarbon emulsion or nanobubbles) or inhibitors of mitochondrial complexes, which may even complement each other, implicating O_2_ re-distribution, facilitating the degradation of HiF, inhibiting DNA repair, and facilitating anti-tumoral immunological response. In essence, mitochondria and the cellular response to hypoxia are two inter-connected systems sensitive to oxygen levels. When HiF-1α is degraded, it promotes mitochondrial metabolism, typically observed in normoxic conditions [32]. Conversely, when HiF-1α is stabilized, it triggers the expression of proteins that, in turn, suppress the mitochondrial metabolism, a characteristic response in a hypoxic condition.

On the other hand, hypoxia is also implicated in the development and severity of numerous age-related and chronic diseases, including cancer and neurodegenerative conditions [5,7,21,22,23]. In the context of cancer, cellular responses to hypoxia play a pivotal role in various aspects of tumor progression, encompassing cancer cell survival, proliferation, epithelial-to-mesenchymal transition (EMT), invasion, angiogenesis, drug resistance, and metastasis [34,35]. Additionally, hypoxic tumors display greater resistance to chemotherapy, radiotherapy, and photodynamic therapy, ultimately resulting in a poorer prognosis and increased patient mortality [34,35]. Please refer to Figure 3B for further insights. Conversely, the central nervous system (CNS) is particularly susceptible to acute hypoxia and can become compromised rapidly, triggering an ischemic cascade. The brain is highly vulnerable to oxidative stress and mitochondrial impairment in hypoxic conditions. Consequently, it experiences neurodegeneration due to energy deficits. Recent research has shed light on significant alterations in mitochondrial balance within the CNS of individuals suffering from chronic neurodegenerative disorders [23,36,37,38,39,40]. As a result, neurodegeneration is characterized by mitochondrial dysfunction, including reduced activity of complex I, cytochrome oxidase, the electron transport chain (ETC), and the occurrence of mitochondrial DNA mutations. This neuroinflammatory degeneration has been extensively studied in the context of conditions such as Parkinson’s disease (PD), Alzheimer’s disease (AD), Huntington’s disease, and hereditary neuropathy [23,36,37,38,39,40]. Herein, as various neurodegenerative diseases share a common association with mitochondrial dysfunction, there is a pressing need to address, repair, and enhance mitochondrial bio-genesis. Within this context, strategies centered on oxygen delivery, such as nanobubbles, have attracted significant attention due to their potential to mitigate hypoxic stress responses and rejuvenate mitochondrial function. To gain a good understanding of the mechanisms and potential applications of nanobubbles in fields like biomedicine, neuroscience, and oncology, it is essential to discover the fundamental principles governing the hypoxia-inducible factor (HiF) pathway and mitochondrial metabolism. Likewise, nanobubbles hold promise in various applications, primarily due to their stability in oxygenation processes, enabling efficient gas exchange within a defined volume. This capacity has the potential to enhance mitochondrial metabolism, alleviate hypoxia, and augment oxygen supply, which is particularly valuable in chronic conditions like cancer, neurodegenerative diseases, and chronic inflammation, amongst others. Herein, to unlock the maximum potential and grasp the mechanism underlying the behavior of OnB in biomedicine, the next section presents foundational principles that govern their bio-performance.

## 3. Nanobubbles in Biomedicine: *Bridging Basic Fundamentals to Practical Application(s)*

Nanobubbles are minuscule gas-filled spherical entities enclosed by an interface (gas–liquid or gas–solid) with diameters typically in the nanometer range, spanning from tens to hundreds of nanometers [8,9]. They represent a distinct class of bubbles, significantly smaller than conventional microbubbles, and are characterized by their unique properties at the nanoscale. In the realm of nanotechnology, nanobubbles have recently garnered considerable attention, mainly attributed to their *exceptional* stability and longevity. This remarkable stability arises from the presence of a thin molecular layer at the gas–liquid interface, which effectively confines the gas within the liquid medium. As a result, nanobubbles exhibit prolonged lifetimes, making them pertinent for biomedicine in various scientific and technological applications, including drug delivery [41,42], gene delivery [43], cancer immunotherapy and chemotherapy [34,35], wound healing, and tissue regeneration, through the intravenous delivery of oxygen carriers such as OnB [44]. Furthermore, nanobubbles have displayed proficiency in surface modification and cleaning at the nanoscale. Their presence can facilitate the detachment of nanosized contaminants or particles from solid surfaces, offering potential applications in surface engineering, nanofabrication, and precision cleaning processes. In the field of medical imaging, nanobubbles serve as contrast agents for advanced ultrasound imaging techniques. Their nanoscale dimensions allow for improved resolution and specificity in visualizing biological tissues, vasculature, and cellular structures. Beyond the biomedical realm, it is perhaps worth mentioning to the interested reader that nanobubbles have promising environmental applications, including groundwater remediation and wastewater treatment, where their ability to facilitate gas transfer and enhance chemical reactions is leveraged to address various environmental challenges. Henceforth, the multi-faceted properties and applications of nanobubbles will continue to be further explored, fueling ongoing research endeavors and innovation across diverse scientific and technological disciplines [41,42,43,44]. 

Various techniques enable the synthesis of nanobubbles, including hydrodynamic cavitation, acoustic cavitation, microfluidics, intense mechanical agitation, and electrolysis, among others. What sets nanobubbles apart from other bubbles are several defining parameters, with bubble diameter being the most critical classification criterion, as outlined by the ISO/TC 281 [45] and ISO 20480-1:2017 [46] standards for terminology and definitions in the R&D&I area of fine bubble technology(ies). The growing interest in nanobubbles across numerous fields stems not only from their expansive surface area and reactivity when compared to macrobubbles and microbubbles but also from their remarkable attributes, including a heightened gas diffusion rate, enhanced cellular uptake, and robust stability against coalescence, collapse, or rupture. These qualities enable nanobubbles to persist in liquid environments for extended periods, spanning several weeks [12,47]. In the ensuing discussion, we delve into the role of physico-chemical features, particularly bubble size, and their interplay with stability. Additionally, we briefly touch upon the significance of factors such as the gas–liquid interface and gas core. Our analysis extends to the implications these factors hold for engineering nanobubbles in diverse bio-medical areas.

### 3.1. Bubble Size and Physico-Chemico-Mechanical Properties

As mentioned previously, the size of nanobubbles, typically falling within the range of 100 to 250 nm, serves as a fundamental parameter that imparts them with inherent and advantageous physico-chemical characteristics [12,47]. This size distribution proves highly advantageous in nanobubble preparations, as it aids in overcoming infiltration limitations and facilitates the disruption of the blood–brain barrier [48]. A noteworthy feature of nanobubbles is their ability to effectively target sites of inflammation, such as those observed in cancer. This efficacy is attributed to the pore cutoff size of the fenestrated endothelium, which ranges from 380 to 780 nm. This mechanism, known as the enhanced permeability and retention effect (EPR effect *basically describes the abnormal characteristics of the blood vessels associated with pathological conditions, particularly in cancer and inflammation*), represents a passive targeting strategy that capitalizes on the efficient accumulation of nanobubbles within tissues featuring elevated vascular permeability [7,15]. To simplify, in the context of EPR (leaky blood vessels and the unique properties of the TME to allow therapeutic agents to accumulate preferentially in cancerous tissue while sparing healthy tissue) and nanobubbles, passive targeting means that nanobubbles, due to their size and other properties, can effectively accumulate in areas of disease, such as tumors or inflamed tissues, without the need for active targeting mechanisms. This passive accumulation can enhance the delivery of therapeutic agents or imaging agents to the specific site of interest, contributing to the effectiveness of various medical interventions. 

Furthermore, their mechanical properties also contribute significantly to their unique characteristics and potential applications. One notable mechanical property is the elevated internal pressure within nanobubbles. Due to their extremely small size, nanobubbles can contain gas at significantly higher pressures compared to larger bubbles. This heightened internal pressure is a result of the Laplace overpressure, which is inversely proportional to the radius of the bubble. Despite this theoretical thermos-dynamic instability, nanobubbles often exhibit impressive stability, remaining intact for extended periods. The mechanical robustness of nanobubbles, particularly their resistance to coalescence, collapse, or rupture, is another noteworthy trait. This resilience against external forces makes them suitable for various applications where durability is crucial. Indeed, understanding the stability of nanosized bubbles can be partially elucidated through their underlying physical behavior, as elaborated below. When any bubble forms within a solution, it creates an interface defined by the surface tension (γ) [49]. The existence of nanobubbles has generated extensive discussion, partly due to the predicted Laplace overpressure associated with them, which can reach several atmospheres, theoretically rendering them thermos-dynamically unstable [49]. However, in contrast to macro- or microbubbles, nanobubbles often exhibit robust persistence characterized by reduced buoyancy, enabling them to remain intact for days, weeks, or even months [12,50]. Additionally, nanobubbles exhibit unique acoustic properties, which have implications in diagnostic imaging and therapeutic ultrasound. Their response to acoustic waves differs from that of microbubbles, opening new opportunities for innovative ultrasound-based techniques. The remarkable attributes defining nanobubbles, including their diminutive size, high surface-to-volume ratio, elevated internal pressure, prolonged stability, electrostatic charge properties, acoustic characteristics, and biocompatibility, have spurred intensive research into their potential across various biomedical applications [51]. In the realm of biomedicine, the combination of nanobubbles’ mechanical attributes, such as their size, stability, and response to external forces, along with their biocompatibility, has led to intensive research into their potential across diverse applications. These mechanical properties, along with their other afore-mentioned physico-chemico-mechanical characteristics, position nanobubbles as a promising platform for advancing various fields, from drug delivery to medical imaging.

### 3.2. Structural Composition and Electrostatic Charge of OnB Affects Gas Core and Diffusion

The gas core of a nanobubble is the central part filled with a gas, typically oxygen or nitrogen [6]. This gas core is encapsulated by a thin layer of liquid, often water, which forms the bubble’s shell. The gas core plays a crucial role in the unique properties and applications of nanobubbles. (**1**) *High internal pressure*: Nanobubbles exhibit significantly higher internal pressure compared to their larger counterparts, such as microbubbles. This increased pressure is a consequence of the Laplace overpressure, which is inversely proportional to the radius of the bubble. The small size of nanobubbles results in elevated internal pressures, which can have implications for their stability and behavior. (**2**) *Gas diffusion*: The gas core of nanobubbles allows for enhanced gas diffusion in liquids. This property is particularly relevant in applications where gas delivery, such as oxygen, to specific biological targets is essential. Nanobubbles can serve as carriers for gases, facilitating their transport to desired locations. The composition of the gas core is another critical factor influencing the stability of nanobubbles [6]. The introduction of gases like perfluorocarbons (PFCs), especially when combined with oxygen, leads to a reduced efflux from the nanobubble, resulting in a prolonged residence time [52]. In parallel research efforts, studies have explored combinations of PFCs, sulfur hexafluoride (SF6), and nitric oxide (NO) with oxygen to formulate micro-nanobubble systems [53,54,55]. On a different note, within the realm of nanodroplets (i.e., *extremely tiny liquid droplets significantly smaller than conventional microdroplets and hence, due to their very small size, can offer more advantages such as increased surface area-to-volume ratios and the ability to penetrate biological barriers or tissues more effectively; they are increasingly being considered as valuable tools in challenging applications including drug delivery, medical imaging, and nanomedicine*), liquid PFCs have found application as O_2_ carriers, capitalizing on their higher O_2_ release capability [50,56].

Nanobubbles have a unique structural composition that sets them apart from larger bubbles. While their precise composition can vary depending on the preparation method and the surrounding medium, the following is a general overview. (**1**) *Gas core*: At the center of a nanobubble is the gas core, typically composed of gases like oxygen, nitrogen, or a mixture of gases. The choice of gas can influence the bubble’s behavior and applications. (**2**) *Liquid shell*: Surrounding the gas core is a thin shell of liquid, which acts as a stabilizing layer. The liquid is often water but can include other liquids or solutions depending on the specific application. (**3**) *Surface molecules*: The interface between the gas core and the liquid shell is defined by surface molecules. These molecules play a critical role in stabilizing the nanobubble and preventing its premature collapse or coalescence. Surface molecules can be surfactants or other stabilizing agents that reduce surface tension. (**4**) *Electrostatic charge*: Nanobubbles may carry a charge on their surface due to interactions with the surrounding medium or the presence of charged molecules. This surface charge can influence behavior, stability, and interactions with other particles and surfaces.

Henceforth, understanding the structural composition of nanobubbles is crucial for customizing their properties to suit specific applications. Researchers and engineers can manipulate these structural elements to craft nanobubbles with tailored characteristics, catering to a wide array of fields like medicine, materials science, and environmental science. For instance, uncoated nanobubbles, lacking a shell structure, have primarily found utility in applications like water treatment and select human applications such as oral drinks and liquid ventilation (Table 1, Table 2, Table 3 and Table 4). These uncoated nanobubbles maintain stability through charge stabilization, interacting with electrolytes in water, ultrapure water, distilled water, or saline solutions. Conversely, most engineered nanobubbles adopt a core–shell structure, where the core predominantly contains the chosen gas content strategically selected for its intended purpose. The shell component exhibits varying compositions and structures, which may include lipids, proteins, polymers, or surfactants. This core–shell configuration has traditionally been employed in the realm of biomedical applications, serving as a contrast agent for ultrasound or as a platform for oxygenation [57]. The outer shell of nanobubbles functions as a protective layer surrounding the gas core, exerting control over crucial characteristics such as stiffness, elasticity, half-life, and the rate of gas exchange. By encapsulating the gas core, the shell effectively prevents direct contact between the gas and the surrounding solvent, minimizing premature interfacial gas diffusion [50,58]. In earlier studies, it was argued that a shell structure was not necessarily a fundamental requirement for bubble formation [11,59]. Consequently, both uncoated and coated nanobubbles or unshelled and shelled nanobubbles have been utilized in the development of therapeutic strategies for various medical applications. However, increased attention has been directed toward coated nanobubbles, primarily because of their ability to incorporate a versatile array of core and shell materials, manipulation options, and bio-conjugation. This versatility extends beyond gases and drugs, allowing for the inclusion of a diverse range of molecules such as proteins, DNA, and ligands [60], enabling precise in vivo targeted delivery (Figure 3C). It is perhaps worth mentioning for the reader herein that shelled micro/nanobubbles (tiny gas-filled bubbles with a protective thin outer shell or coating that is typically composed of various materials, including lipids, proteins, polymers, surfactants, or other biocompatible substances) exhibit a more gradual external diffusion of internal gases [7,58,61,62,63], contributing their unique characteristics.

## 4. NanoBubbles as a Platform for O_2_ Delivery: *Innovative OnB-Mediated Oxygenation*

In the ever-evolving landscape of medical science and biotechnology, the emergence of nanobubbles as a platform for oxygen delivery has sparked innovation and excitement. These minuscule nano-scaled bubbles have paved the way for groundbreaking approaches to oxygenation, offering a promising solution to address critical challenges in various fields, from medicine to environmental science. Indeed, the rising incorporation of nanobubbles into biomedicine is underpinned by a range of highly beneficial characteristics, as detailed above. To re-emphasize, these *versatile* physical–chemical–mechanical properties encompass stability, the ability to provide a larger surface area for interaction, the availability of uniform or varied size distributions, increased payload capacity, efficient interfacial gas diffusion for precise gas delivery, biodegradability, biocompatibility, accelerated cellular uptake, and the strategic utilization of the EPR effect. The potential of harnessing these nanobubble platforms within clinical settings to encapsulate and deliver oxygen with exceptional precision (targeted oxygen delivery) is increasingly feasible. Indeed, their diminutive size allows them to navigate intricate biological pathways and reach specific cellular or tissue targets with unparalleled efficiency. These advancements are driven by the development of enhanced formulations of stable and versatile OnB that can be tailored to a myriad of applications, ranging from enhancing oxygen levels in biological systems to serving as contrast agents in medical imaging. Development and innovation efforts continue to advance the practical integration of nanobubbles into clinical application, which is becoming increasingly plausible with the translation of novel formulations, including coated and core–shell nanobubbles, which have been and are being developed, including in our labs, to enhance gas solubility and extend oxygen retention durations, all while avoiding the formation of problematic macro-bubbles [12,47]. In this era of innovation, the exploration of OnB as a platform for oxygen delivery represents a promising frontier. Their ability to navigate the complexities of biological systems and precisely deliver oxygen opens new avenues for addressing hypoxia-related conditions, improving medical diagnostics, and revolutionizing drug delivery and release strategies. As we delve deeper into the realm of OnB, their potential to re-shape the landscape of oxygenation in various fields is an exciting prospect with immense promise for the future of biomedicine. 

**O_2_ release strategies involving OnB technologies**: The efficient release of oxygen from nanobubbles into a liquid is a spontaneous process driven by the pre-existing concentration gradient between the gas core and the surrounding liquid phase. Moreover, OnB serves a dual purpose as contrast-enhancement agents for ultrasound (US) imaging, offering the distinct advantage of enabling guided and real-time monitoring of oxygen delivery [53]. In scenarios involving an ultra-sound (US) -triggered release mechanism, the emission of high-intensity US waves gives rise to distinct high and low-pressure resonance zones within the propagating wave. These zones result from resonant phenomena, ultimately leading to the disruption of bubbles and the subsequent liberation of the encapsulated gas core. We present a current summary of studies and potential applications of both uncoated and coated OnB in the field of biomedicine (Table 1, Table 2, Table 3 and Table 4). The intravascular (IV) delivery of oxygen using OnB has emerged as a prominent administration route in cancer therapies and for conditions characterized by microcirculation dysfunction [44]. Additionally, OnB shows promise as a secure platform for blood oxygenation, achieving oxygen saturation levels of up to 95% without activating the blood complement system or causing hemolysis. Moreover, their minimal IV fluid injection volume requirement underscores their practicality and feasibility [6]. Research indicates that the cellular uptake of OnB involves complex endocytosis processes, thus preserving cellular functionality and supporting the achievement of desired therapeutic outcomes [93]. In essence, oxygen release from OnB can occur through gentle methods such as US cavitation or diffusion mechanisms. While alternative oxygen release mechanisms have been explored, such as those involving OnB assisted by photodynamic processes [33] or pH variations [69], a significant gap persists in our molecular understanding of how OnB behave within cells, extending to the intricate processes by which mitochondria and HiF-α sense OnB presence.

## 5. Molecular Insights into the Mechanism of OnB in Chronic Diseases and Disorders

Chronic diseases, ranging from neurodegenerative disorders to cancer, pose significant challenges to both patients and healthcare providers. Hypoxia, a condition characterized by reduced oxygen levels in tissues, plays a pivotal role in the progression and severity of these diseases. Understanding the molecular mechanisms involved in hypoxia-related conditions and the emerging role of OnB is shedding light on innovative therapeutic strategies. Indeed, the exploration and study of the mechanisms underlying the impact of OnB in biomedical contexts has spurred numerous investigations at the cellular and molecular levels. Presently, our understanding of the biological consequences of OnB is most comprehensive in studies related to neurodegenerative disorders and cancer. OnB have been linked to the enhancement of mitochondrial activity, the activation of phosphatidylinositol-3-kinase (PI3K), and the reversal of hypoxic cell responses [7,15]. (**1**) *Oxygen delivery*: OnB function as carriers of oxygen, releasing it through various mechanisms, including diffusion, ultrasound-induced cavitation, photodynamic processes, or pH-responsive triggers. This controlled oxygen release is critical for replenishing oxygen-deprived tissues. (**2**) *Mitochondrial regulation*: Mitochondria, the cellular powerhouses, are profoundly affected by oxygen levels. Reduced oxygen availability triggers a metabolic crisis marked by mitochondrial dysregulation. OnB can provide oxygen directly to mitochondria, supporting their function and cellular metabolism. (3) *Hypoxia-inducible factor* (*HiF*) *pathway*: HiF is a key transcription factor that orchestrates cellular responses to low oxygen levels. OnB can modulate the HiF pathway, affecting the expression of genes involved in metabolism, glycolysis, and mitochondrial function. This interplay between OnB and HiF offers insights into how oxygen nanobubbles impact cellular responses in hypoxic conditions. (**4**) *Targeted oxygenation*: OnB can navigate complex biological environments to reach specific cells or tissues, providing localized oxygenation. This targeted delivery approach minimizes potential side effects and enhances therapeutic O_2_ precision. To summarize, the molecular mechanisms of OnB in chronic diseases involve precise oxygen delivery, regulation of mitochondrial function, and modulation of the HiF pathway. These mechanisms offer promising avenues for therapeutic interventions in a wide range of chronic diseases, with the potential to improve patient (clinical) outcomes and therefore, quality of life. Ongoing R&D&I or research, development and innovation aim to further elucidate these underlying molecular mechanisms and optimize OnB-based therapies for their clinical use.

### 5.1. OnB and Neurodegenerative Disorders/Diseases

In conditions like Alzheimer’s and Parkinson’s disease, where mitochondrial dysfunction and hypoxia play significant roles, OnB present promising therapeutic interventions by bolstering neuronal health and function. Mitochondria, pivotal organelles in cellular function, play a central role in energy metabolism through the generation of ATP via oxidative phosphorylation (OXPHOS). Maintaining tissue homeostasis demands substantial energy across various cell types, exemplified by the crucial role of oligodendrocytes (OL) in upholding myelin integrity, which entails significant metabolic activity and ATP consumption. Exposure to OnB results in heightened mitochondrial activity in both neurons and glial cells, marked by increased levels of PGC1α. This enhancement in mitochondrial dynamics translates to amplified respiratory capacity, ATP production, and augmented mitochondrial content in terms of cell size and area (as illustrated in Figure 4A–E) [21]. Similar outcomes have been observed in various other cell types, indicating an improved mitochondrial bioenergetic state [37,94]. Neurons treated with OnB exhibit a notable increase in mitochondrial ATP production compared to untreated cells. Significantly, blocking ATP synthesis nullifies the positive effects of OnB. Notably, even mere exposure to OnB has shown the remarkable ability to enhance the survival rates of neurons and oligodendrocytes by actively suppressing apoptosis through the downregulation of caspase 3/7 activity, even under conditions of metabolic stress such as glucose deprivation (Figure 4F,G) [5]. This phenomenon is characterized by a higher oxygen consumption rate (OCR), confirming that OnB primarily enhances mitochondrial bio-energetic state under both basal and stress-induced conditions (Figure 4H) [5]. For instance, the measured basal OCR in neurons was higher in the presence of OnB-RNS60 compared to an optimal glucose-containing medium. The correlation of OCR with mitochondrial respiration and extracellular acidification rate (ECAR) with glycolysis reveals a pronounced upward and rightward shift in the presence of RNS60 under glucose-deprived medium conditions (Figure 4H) [5]. This observed shift signifies simultaneous enhancements in both oxidative phosphorylation and glycolysis. Alignment with the shift in oxidative phosphorylation corroborates the modulatory influence exerted by this type of un-coated OnB, which has been shown to modify the electro-physiological properties of the *Xenopus laevis* oocyte membrane (cellular/plasma membrane of oocytes or immature ovarian cells from the African clawed frog, *Xenopus laevis*) through increasing mitochondrial-based ATP synthesis (process by which cells generate adenosine triphosphate, a molecule that stores and releases energy for cellular activities, mainly via cellular respiration) [95]. 

These findings highlight the potential of OnB to enhance neuronal survival and function, particularly in the face of metabolic stress conditions, potentially serving as a source of fuel for mitochondrial-related ATP processes [94]. In addition to the outlined benefits and mechanisms, it is worth mentioning some additional important points related to the use of OnB in addressing mitochondrial dysfunction and hypoxia-related conditions. (**1**) *Reduced oxidative stress*: OnB-mediated oxygenation can help mitigate oxidative stress, a common feature in neurodegenerative diseases. By enhancing mitochondrial function and ATP production, OnB may indirectly reduce the production of harmful reactive oxygen species (ROS), thus contributing to neuroprotection. (**2**) *Neuro-inflammation modulation*: OnB treatment has been associated with the suppression of neuroinflammatory responses. Mitochondrial dysfunction and oxidative stress often trigger inflammation in neurodegenerative conditions. The OnB ability to improve mitochondrial health can help dampen neuroinflammation, potentially slowing down disease progression. (**3**) *Enhanced drug delivery*: OnB can serve as carriers for various therapeutic agents, including drugs and gene therapies. This capability allows for targeted drug delivery to specific cell types or tissues affected by neurodegenerative diseases. It can improve the efficacy of treatments while minimizing off-target effects. (**4**) *Potential for combination therapies*: OnB-based interventions can be combined with other treatment modalities, such as photodynamic therapy or immunotherapy, to create synergistic effects in combating neurodegenerative diseases. These multimodal approaches hold promise for more comprehensive disease management. (**5**) *Non-invasive delivery*: OnB can be administered via non-invasive routes like intravenous injection, minimizing the need for invasive procedures. This characteristic is advantageous, especially for the elderly or geriatric patients and/or those with advanced stages of neurodegenerative diseases and/or institutionalized. (**6**) *Clinical translation*: While much R&D&I remains to be discovered and conducted, the potential clinical translation of OnB-based therapies offers ample hope for improving the quality of life for individual patients affected by neurodegenerative diseases. Further pre-clinical and clinical investigations are required to validate the bio-safety and bio-efficacy of OnB interventions.

#### 5.1.1. Role of OnB in Increasing/Enhancing Mitochondrial Bio-Genesis and Cellular Energy

Mitochondrial bio-genesis refers to the process by which new mitochondria are formed within cells, and it plays a crucial role in maintaining cellular energy production and overall cellular health. Cellular metabolism is a controlled biological process that involves numerous signaling pathways and transcriptional networks. As elucidated earlier, PGC1α is a pivotal orchestrator governing metabolism and mitochondrial bio-genesis [21]. Functioning as a transcriptional factor, PGC1α serves as a transcriptional factor that links external stimuli with mitochondrial metabolism [96]. However, the mechanism underpinning PGC1α activation and the subsequent upregulation of mitochondrial bio-genesis remains unclear. Nanobubble-based O_2_ delivery systems act as an alternative strategy to not only amplify ATP production but also to augment mitochondrial size and mitogenesis through a series of mechanisms, encompassing the following two main actions [21,96]:OnB stimulate mitochondrial bio-genesis by inducing the upregulation of PGC1α (Figure 5).OnB upregulate the expression of nuclear respiratory factor 1 (NRF1), mitochondrial transcription factor A (TFAM), and genes closely associated with mitochondrial bio-genesis (Figure 5).

To recap, OnB may contribute to increasing mitochondrial bio-genesis via the following mechanisms. (**1**) *Enhanced oxygen supply*: OnB serve as a source of supplemental oxygen to cells, especially in oxygen-deprived or hypoxic environments. Mitochondria are the primary sites for cellular respiration, where oxygen is used to generate ATP (adenosine triphosphate), the cell’s energy currency. By providing additional oxygen, OnB can help ensure that mitochondria have an adequate oxygen supply for efficient ATP production. (**2**) *Activation of cellular signaling pathways*: OnB treatment has been associated with the activation of cellular signaling pathways that regulate mitochondrial bio-genesis. One key player in this process is peroxisome proliferator-activated receptor gamma coactivator 1-alpha (PGC1α), a master regulator of mitochondrial bio-genesis. OnB treatment may increase the expression and activity of PGC1α, leading to the synthesis of new mitochondria. (**3**) *Improved respiratory capacity*: OnB-induced oxygenation can enhance the respiratory capacity of mitochondria. Mitochondrial bio-genesis involves not only the formation of new mitochondria but also the optimization of the function of the existing mitochondria. OnB may promote the efficient use of oxygen in the electron transport chain, further boosting ATP production. (**4**) *Neuroprotection*: In conditions like neurodegenerative diseases, where mitochondrial dysfunction is a hallmark, the ability of OnB to increase mitochondrial bio-genesis is of particular importance. Mitochondrial health is crucial for neuronal survival and function. OnB-induced mitochondrial bio-genesis may help protect neurons from degeneration and maintain their energy demands. (**5**) *Potential therapeutic applications*: Harnessing OnB to promote mitochondrial bio-genesis (i.e., highly regulated process by which cells increase the number and mass of their mitochondria via the synthesis and incorporation of new mitochondrial components, such as proteins, lipids, and DNA, to expand the existing mitochondrial population; deemed key for maintaining cellular energy homeostasis, adapting to metabolic demands, and ensuring proper functioning of cells) holds substantial promise for the design and development of novel therapeutic interventions. Henceforth, by supporting mitochondrial health and function, OnB-based treatments may benefit individuals diagnosed with and suffering from chronic and severe conditions ranging from neurodegenerative to cardiovascular disorders to cancer.

#### 5.1.2. Mitochondrial Bio-Genesis via the Phosphatidylinositol 3-Kinase (PI3K) Enzyme

Beyond the promotion of mitochondrial growth by OnB via the increased expression of PGC1α [21], emerging scientific evidence underscores the pivotal role of a key signaling molecule, namely PI3K, in mediating numerous beneficial effects attributed to these nanobubbles [36,97]. PI3K is a vital signaling enzyme that governs a broad spectrum of biological responses, encompassing cell proliferation, cell viability, and programmed cell death, known as apoptosis [98]. Class IA PI3K consists of a regulatory 85-kDa subunit and a catalytic 110-kDa subunit (p85:p110α/β), regulated by multiple receptor tyrosine kinases (RTK) and G protein-coupled receptors (GPCR). Conversely, class IB PI3K comprises a dimer consisting of a 101-kDa regulatory subunit and a p110γ catalytic subunit (p101/p110γ). Recent investigations have corroborated the hypothesis that OnB might also induce heightened mitochondrial bio-genesis in neurons by activating PI3K and engaging transcriptional co-activators (Figure 5) such as CREB (*knowingly to be induced by a variety of growth factors, cytokines and inflammatory signals prior to mediating the transcription of genes*). 

The activation of PI3K has been closely linked to the suppression of pro-inflammatory responses in microglia, achieved through the downregulation of nuclear factor-Κb [97]. Furthermore, it has been implicated in enhancing T regulatory cell activity, resulting in reduced nitric oxide production, and subsequently restraining the differentiation of Th17 and Th1 cells [36]. Consequently, the effects of OnB observed in T cells, glial cells, or neurons can be inhibited by blocking the PI3K pathway [21,36,78]. Nevertheless, several significant questions remain actively under investigation, including the precise mechanism through which charge-stabilized nanobubbles activate PI3K and the identification of other downstream pathways contributing to this activation. PGC1α exerts regulatory control over various mitochondrial functions and essential mitochondrial transcription factors like TFAM. PGC1α’s activity is subject to direct modulation through various post-translational modifications, including acetylation, methylation, ubiquitination, and phosphorylation [99]. However, the underlying mechanisms governing the expression and functional regulation of PGC1α, particularly its intricate interactions with proteins such as SIRT, AMPK, MAPK, and Akt, remain incompletely understood [99]. In parallel, the multifaceted role of TFAM in terms of its impact on mtDNA, the mitochondrial network (including mitophagy, mito-fission, and mito-fusion), and functional assays for mitochondria remains a relatively unexplored territory. Conversely, the upregulation of PGC1α is indirectly controlled by a mechanism involving the expression of NRF1 and CREB [99,100]. Recent studies conducted on dopaminergic neuronal cells and glial cells have shed light on the role of Akt, a downstream effector of PI3K, in orchestrating the activation of CREB in the context of OnB treatment [21,37]. These findings suggest a model in which OnB increase the expression of PGC1α through class IA PI3K-mediated activation of CREB, subsequently leading to enhanced mitochondrial bio-genesis. Notably, experiments utilizing siRNA knockdown targeting both PGC1α and CREB or the inhibition of PI3K have demonstrated the blockade of OnB-mediated expression of both TFAM and NRF1 [21,37]. 

To recap the critical importance of regulating mitochondrial bio-genesis for maintaining cellular energy homeostasis and responding to metabolic demands, the following summarizes how PI3K is involved in the process. (**1**) *Activation of PI3K*: Phosphatidylinositol 3-kinase (PI3K) is an enzyme that phosphorylates the lipid molecule phosphatidylinositol 4,5-bisphosphate (PIP2) to generate phosphatidylinositol 3,4,5-trisphosphate (PIP3). This lipid modification triggers a cascade of downstream signaling events. (**2**) *Activation of Akt* (*Protein Kinase B*): PIP3 generated by PI3K activates a protein kinase called Akt or Protein Kinase B (PKB). Akt is a central player in cell signaling pathways that regulate cell growth, survival, and metabolism. (**3**) *Akt-mediated signaling*: Activated Akt can phosphorylate and regulate various downstream targets, including transcription factors and other proteins involved in mitochondrial bio-genesis. One of the key targets is the transcriptional coactivator PGC-1α (Peroxisome proliferator-activated receptor gamma coactivator 1-alpha). (**4**) *Role of PGC-1α*: PGC-1α is a master regulator of mitochondrial bio-genesis. When activated by Akt, PGC-1α promotes the expression of genes responsible for mitochondrial DNA replication, transcription, and protein synthesis. It also enhances the formation of new mitochondria by coordinating the synthesis of mitochondrial components. (**5**) *Increased mitochondrial mass*: The activation of PI3K-Akt signaling and subsequent phosphorylation of PGC-1α result in the increased expression of mitochondrial genes and the growth of existing mitochondria. This leads to an overall increase in mitochondrial mass and function. (**6**) *Enhanced cellular energy production*: As mitochondrial bio-genesis progresses, the cell becomes more equipped to produce ATP through oxidative phosphorylation, a process that occurs within the mitochondria. This enhanced ATP production supports the cell’s energy needs and metabolic processes. (**7**) *Cellular adaptation*: The activation of PI3K-Akt signaling and mitochondrial bio-genesis can be a response to various cellular stimuli, including changes in energy demands, exercise, and environmental factors. Cells tend to adapt to distinct metabolic needs through modulating their mitochondrial content. 

Henceforth, the PI3K–Akt pathway and its involvement in mitochondrial bio-genesis have significant implications in various physiological and pathological conditions. Dysregulation of this pathway can contribute to metabolic disorders, neurodegenerative diseases, cancer, and other health-related issues. Understanding the molecular mechanisms underlying mitochondrial bio-genesis via PI3K signaling provides valuable insights into cellular physiology and the development of potential therapeutic interventions for diseases characterized by mitochondrial dysfunction. It is crucial to acknowledge the complexity of transcriptional regulation and its multi-faceted role. Henceforth, further discovery and experimental R&D&I studies are desirable to comprehensively understand the intricacies of this transcriptional regulation and its broader effects on the various mitochondrial processes, including the electron transport chain, oxidative stress response, ROS detoxification, tricarboxylic acid cycle or TCA, and mitochondrial bio-genesis regulation.

### 5.2. OnB and Overcoming Hypoxia and Hypoxic Conditions

In a normoxic environment (conditions characterized by normal, standard, or adequate levels of oxygen and oxygen concentration), human cells predominantly allocate more than 90% of the available oxygen to mitochondrial respiration [32]. This leaves approximately 10% of the oxygen in the cytosol, which is adequate for triggering the degradation of HiF-1α. However, under sustained or prolonged hypoxic conditions, mitochondria intensify their oxygen consumption, nearly depleting the cytosolic oxygen pool and consequently stabilizing HiF-1α [32]. Once activated, HiF-1α plays a pivotal role in the transcription of numerous genes, including TGF-α, VEGF and VEGF Receptor FLT1, GLUT1, MMPs, EPO, HK2, CA9, and iNOS [32]. In recent years, considerable attention has been focused on unraveling the complex role of HiF-1α in the context of cancer. The upregulation and stabilization of HiF-1α within cancer cells have been closely associated with increased tumor and cell survival, heightened angiogenesis, enhanced proliferation of cancer cells, and elevated resistance to both chemotherapy and radiation therapies [64,101]. Modern research efforts have shed light on the significant influence of OnB on HiF-1α dynamics. These nanobubbles have exhibited the ability not only to inhibit HiF-1α expression [65] but also to substantially reduce hypoxia-induced resistance to radiation in various cancer cell lines, such as EBC-1 and MDA-MB-231 cells (human lung and breast cancer cells) [64]. Significantly, OnB have shown the potential to serve as a sensitizing adjuvant when administered in combination with low-dose radiation therapy, potentially enhancing treatment bio-efficacy in cancer patients without increasing toxicity. This is due to the ability of OnB in helping to increase O_2_ availability within the tumor microenvironment (tumors often have regions of low oxygen or hypoxia, which can make them more resistant to radiation therapy, and radiation therapy relies on the presence of oxygen to generate free radicals and induce DNA damage in cancer cells; hence, when tumor cells are hypoxic, radiation therapy becomes less effective). Therefore, by delivering oxygen directly to the tumor site, OnB can potentially alleviate tumor hypoxia, making cancer cells more susceptible to the damaging effects of radiation, thereby enhancing its efficacy.

#### 5.2.1. Molecular Bases to Downregulate/Suppress HiF-1α *Post*-OnB Application/Therapy

OnB are designed to release oxygen when they reach the hypoxic regions within tissues or tumors. Such localized oxygen delivery can increase the oxygen tension in these areas, thereby reducing the stabilization of HiF-1α (sensitive to oxygen levels), as well as in normoxic conditions undergoing both hydroxylation and proteasomal bio-degradation. 

Hence, HiF-1α serves as a crucial physiological indicator of oxygen delivery and is studied as the primary physiological probe for oxygen delivery due to its rapid degradation in the presence of oxygen. Both in vitro and in vivo studies have consistently demonstrated a reduction in HiF-1α expression following OnB exposure [7,72,91]. When tumor cells are exposed to OnB in vitro, the hypoxic conditions are reversed, accompanied by the suppression of HiF-1α levels (Figure 6A–C) [7]. Similarly, in a mouse model, a significant decrease of 75% in mRNA levels and 25% in protein levels of HiF-1α was observed (Figure 6D–F) [65]. These oxygen-induced changes within tumors have also led to a notable reduction in VEGF expression in tumors treated with the oxygen nanobubble suspension. Furthermore, lung cancer and breast cancer cells have exhibited clear suppression of hypoxia-induced HiF-1α expression when cultured in media containing OnB compared to cells cultured in distilled water-based media [64,65]. Herein, remember that OnB, while reducing ROS levels within hypoxic regions and interfering with the signaling pathways that activate HiF-1α under hypoxic conditions, might inhibit the activation of receptor tyrosine kinases (RTKs) or other upstream regulators involved in HiF-1α stabilization. Further, OnB treatment may activate cellular pathways that antagonize HiF-1α expression or stability. These pathways could involve negative feedback loops, post-translational modifications, or competitive binding to HiF-1α. Additionally, OnB could induce apoptosis (programmed cell death) in hypoxic cells, which may lead to the degradation of HiF-1α. Apoptosis can be triggered in response to prolonged hypoxia and is associated with the degradation of various cellular proteins, including HiF-1α. Finally, to re-emphasize, OnB may indirectly contribute to the suppression of HiF-1α by locally reducing levels_ROS_.

#### 5.2.2. OnB and Epigenetic Modulation in Tumors

Briefly, epigenetic modifications are heritable changes in gene expression that do not involve alterations in the underlying DNA sequence but can influence gene activity. In the context of neoplasia, where tumoral cells exhibit uncontrolled proliferation and insufficient angiogenesis, the tumor’s microenvironment often becomes hypoxic. In response to this prevailing hypoxia, various oncogenic mechanisms come into play, with epigenetic modifications taking a prominent role in promoting increased tumor growth and enhancing the survival of cancer cells [102]. These epigenetic changes encompass a wide spectrum of alterations, including hypomethylation, primarily activating oncogenes [103], gene-specific hypermethylation of CpG islands in the promoter regions of tumor suppressor genes, rendering them inactive, and contributing to aberrant cell proliferation [104]. Central to these processes are the ten–eleven translocation (TET) enzymes, a class of dioxygenases dependent on Fe^2+^ and α-ketoglutarate. These enzymes facilitate the conversion of 5 mC to 5 hmC and its downstream derivatives [105,106]. The quantification of 5 mC levels presents a clear and statistically significant reduction (α = 0.05) in DNA methylation within hypoxic cells when exposed to OnB, as compared to control cells (Figure 6G) [72]. This observation is further supported and validated through enzyme immunoassay (EIA) and liquid chromatography–mass spectrometry (LC-MS/MS) (Figure 6G,H) analyses [72]. OnB demonstrate their capacity to effectively reprogram crucial genes associated with hypoxia and tumor suppression, such as MAT2A and PDK-1. Furthermore, a notable negative correlation emerges between the expression levels of PDK-1 and MAT2A and both tumor size and metastasis [72]. Concurrently, the methylation status of a panel of 22 tumor suppressor genes was meticulously profiled [72]. Remarkably, treatment with nanobubbles leads to a significant reduction in promoter methylation for 11 of these genes (Figure 6G). This intriguing observation suggests that OnB have the potential to reverse hypoxia-induced epigenetic changes, steering the cancerous DNA methylation landscape toward a more normalized state. This process entails both global re-methylation and the selective demethylation of specific tumor suppressor genes. A notable example is the reactivation of cell cycle regulation mediated by genes such as p57 (CDKN1C), p21 (CDKN2A), and p73 (TRP73) upon nanobubble treatment. This observation suggests that the reversal of hypoxia by nanobubble treatment has the potential to reshape the cancerous DNA methylation landscape towards a relatively normal status. It involves global re-methylation and the targeted demethylation of specific tumor suppressor genes. Notably, nanobubble treatment leads to the reactivation of essential cell cycle control mechanisms mediated by p57 (CDKN1C), p21 (CDKN2A), and p73 (TRP73). These genes play crucial roles in suppressing abnormal cell proliferation and inducing apoptosis in cancers [107,108]. Importantly, their functions are often silenced through aberrant hypermethylation across a range of various cancer types [109,110]. Consequently, the epigenetic modulation of these genes using OnB may contribute to the observed inhibition of tumor progression in vivo.

To recap, while the specific mechanisms of how OnB may modulate epigenetic changes in tumors are not yet fully understood, some of the potential ways in which OnB could impact epigenetics include the following. (**1**) *DNA methylation*: OnB could potentially influence DNA methylation patterns in tumor cells. DNA methylation involves the addition of methyl groups to DNA, often resulting in gene silencing. Changes in oxygen levels can affect the activity of DNA methyltransferases, enzymes responsible for DNA methylation. OnB may influence the activity of these enzymes, thereby altering DNA methylation patterns. (**2**) *Histone modifications*: OnB might affect histone modifications, which are chemical changes to histone proteins around which DNA is wrapped. These modifications can influence chromatin structure and gene expression. Changes in oxygen levels can impact the activity of enzymes that add or remove these chemical groups from histones, and OnB could potentially play a role in this process. (**3**) *Non-coding RNAs*: OnB could influence the expression of non-coding RNAs, such as microRNAs and long non-coding RNAs, which are known to have epigenetic regulatory functions. Changes in oxygen levels within tumor cells might affect the production and function of these non-coding RNAs, leading to epigenetic changes. (**4**) *Cellular signaling pathways*: OnB have been shown to influence cellular signaling pathways, including those involved in cell survival, proliferation, and apoptosis. These pathways can intersect with epigenetic regulatory mechanisms, and changes in signaling pathways induced by OnB may contribute to epigenetic alterations in tumor cells. Finally, remember that the precise mechanisms through which OnB modulate epigenetic changes in tumors are an active area of research, and more studies are required to fully understand these processes. Epigenetic modifications, represent a multi-faceted layer of regulation with substantial impacts on gene expression and cellular behavior, henceforth, are complex and can have profound effects on gene expression and cellular behavior; thereby, uncovering the specific epigenetic consequences and nature of changes following OnB therapy in tumors is an intriguing avenue of investigation.

## 6. Other Remarks

The utilization of high-weight gases, exemplified by perfluorocarbon (PFC) emulsions or nanobubbles formulated with an oxygen gas core, has demonstrated enhanced oxygenation. This advantage is manifested through their elevated gas diffusion rate, superior cellular uptake, and robust stability against coalescence, collapse, or bursting, enabling their sustained viability in liquids over prolonged durations [12,47]. Within the biomedical domain, this holds particular significance for addressing hypoxic regions within tumors, non-healing wounds, or tissues affected by mitochondrial dysfunction. However, it is noteworthy that while various studies have extensively explored applications, there is a discernible paucity of research delving into the intricate biological mechanisms after cell exposure to these engineered nano-entities. Several reports indicate that upon OnB administration, cellular uptake involves intricate endocytosis processes, where they are encapsulated within membranous bag-like structures [93]. While this is proposed as the primary mechanism of internalization, preserving cellular functionality, and supporting the attainment of desired therapeutic outcomes [93], an alternative perspective posits that depending on the electrostatic charge, OnB may traverse the cell membrane through channels (passive diffusion), gaining direct access to the cytosol. This aspect holds significance for a more comprehensive understanding of O_2_ sensing by mitochondrial consumption [32] or cytoplasmic enzymes (*hydroxyprolines*) acting as *key* modulators of HiF-1 alpha [30]. 

Regarding underlying cellular and molecular mechanisms, our focus in this article is directed towards specific groups of diseases or medical conditions wherein nanobubble-mediated oxygenation exhibits and has exhibited promise. Notably, the detrimental effect of metabolic stress is present in chronic diseases such as MS or multiple sclerosis, Alzheimer’s, and Parkinson’s disease, amongst others (including cardiovascular, respiratory, ischemia/ischemic micro-environment, and/or cancer). In these scenarios, the role of OnB as modulators of respiration involves the activation of PI3K-Akt signaling, instigating mitochondrial bio-genesis via PGC1α [21,36,78]. This cascade results in enhanced neurological survival, accompanied by an anti-inflammatory effect observed in T-regs. However, in other cell types with distinct metabolic demands, such as the hypoxic tumoral microenvironment (TME), chronic wounds, or cardiovascular diseases, it remains unclear if the effects of OnB treatment are also elucidated through mitochondrial bio-genesis via PI3K/CREB and PGC1α. Whilst the studies cited above succeeded in hypoxia reversal or normoxic condition with metabolite or via FBS deprivation conducted on 2D cell monolayers, there is a lack of evidence for a better model to mimic in vivo conditions, allowing for a more accurate exploration of metabolic alterations and hypoxic environment. In addition, our comprehensive literature review concerning cancer, amongst other indications, underscores a notable deficiency in studies that specifically address the diverse metabolic requirements within 3D models. These models, such as spheroids or cell constructs characterized by their 3D architecture, offer a more physiologically relevant representation of cellular environments when compared to traditional 2D models. Regardless of the literature available for nanobubble-mediated oxygenation and its molecular crosstalk with mitochondria [21,36,78], there is still a lack of studies directly correlating the release of oxygen with mitochondrial content or mitochondrial fitness, along with the assessment of ROS levels and the maintenance of mitochondrial membrane potential (Δψm). This gap in research is particularly notable in areas such as cancer, chronic wounds, or cardiovascular and respiratory diseases, with neuroscience being one of the main fields of focus seeking the understanding of the underlying molecular mechanism. We believe that addressing such knowledge gaps could contribute significantly to advancing our understanding of the intricate interplay between nanobubble-mediated oxygenation and cellular processes in diverse pathological contexts. Focusing specifically on tumors, these OnB have demonstrated efficacy in reversing hypoxic conditions, leading to a subsequent reduction in the expression of HiF-1α across various tumor types and cancer cell lines (Table 1). For instance, Khan et al. [7] synthesized OnB through the sonication of an oxygen-saturated lipid solution, resulting in reduced HiF-1α expression in MDA-MB-231 cells cultured in a hypoxic environment. This line of investigation was extended in a subsequent study [15], where OnB were loaded with anti-tumoral drugs, such as Doxorubicin (Dox), to achieve an enhanced therapeutic effect. Notably, both O_2_ and Dox delivery in these studies relied on a passive release mechanism, suggesting the need for an external ultrasound (US) trigger. In contrast, the field of cancer research has also witnessed significant benefits and developments in utilizing OnB as contrast agents, elevating nanobubbles to the status of acoustic-responsive agents for molecular imaging and theragnosis. Furthermore, this field is evolving by incorporating diverse stimuli to modulate release mechanisms. Recent approaches include ultrasound [53], high-intensity focused ultrasound (HIFU) [48], and pH-responsive nanobubble platforms [69]. These advancements signify a broader exploration of stimuli-responsive strategies for precise control over gas and drug release, showcasing the potential versatility of OnB use in the biomedical realm. 

Finally, it is perhaps worth mentioning herein that among the diverse nano-entities and nano-scaled systems utilized in engineering nano-theragnostic platforms, a plethora of micro and nano-particles (NPs) have been developed to exploit various release triggers. Noteworthy examples include gold NPs (in response to infra-red radiation) [111], mesoporous silica NPs (pH) [112], super-paramagnetic iron oxide NPs (magneto-thermal) [113], micro-/nano-bubbles [114], nano-droplets [115], and echogenic liposomes [116] (responsive to ultrasound). Consequently, interventions based on nanobubbles which translate into enhanced echogenic properties, can also be integrated into platforms alongside other nano-entities, such as liposomes [42]. This integration enables the application of diverse treatment modalities, including photodynamic therapy (PDT) [117], photothermal therapy (PTT) [118], chemotherapy, gene therapy [119], or immunotherapy [120], fostering synergistic effects in combatting cancer and enhancing efficacy in other conditions associated with hypoxia and mitochondrial dysfunction (Table 1). Yet, it is imperative to note that nanobubble-mediated oxygenation appears to grapple with challenges related to pre-mature O_2_ release alongside storage-related issues [18,121,122,123], limitations posing a significant hurdle, potentially diminishing efficacy in targeting and delivering gas/drugs effectively.

## 7. Conclusions

The use of ultrafine bubbles as customizable nanocarriers or platforms infused with gas/drug-based nanotechnology represents a highly promising frontier, particularly in the context of conditions related to oxygen deficiency, such as neurodegenerative disorders, cardiovascular diseases, and cancer. Nanobubbles offer a range of advantages, including their stability for oxygenation, efficient gas exchange within confined volumes, and prolonged gas delivery compared to conventional micro/macro bubbles. These attributes encompass negative electric surface charge density, extended lifespan, reduced buoyancy, the ability to carry multiple gases, and strong echogenicity. Both coated and uncoated OnB have shown significant potential in enhancing mitochondrial metabolism, overcoming hypoxia, and increasing oxygen availability across various diseases, including cancer, neurodegenerative disorders, and chronic inflammation. Coated nanobubbles have garnered specific attention for their versatility in carrying not only gases but also various molecules, including drugs, proteins, DNA, and ligands, for precise in vivo delivery. While existing evidence mainly consists of in vitro studies highlighting the robustness of OnB in cellular oxygenation and their ability to downregulate HiF-1α while enhancing mitochondrial function through PI3K-mediated increases in PGC1α and TFAM transcription, several aspects require further exploration. These include the measurement of oxygen dynamics within cells, investigations into antioxidant cell signaling (such as SOD2), assessments of mitochondrial health and fitness, and comprehensive evaluations of biosafety and targeting precision in preclinical and clinical trials, especially for intravenous OnB applications. Additionally, uncharted territories such as muscular recovery, cutaneous lesion oxygenation, and tissue regeneration offer exciting prospects for future research. Although some lipid-shelled microbubbles (1–10 μm, OmB) are already in widespread clinical use as contrast agents for echocardiography, concerning the translational barriers associated with nanobubble-mediated oxygenation in cancer treatment and other chronic diseases, these methods appear to grapple some pivotal challenges. Firstly, there is an issue with pre-mature oxygen release and storage inadequacies, wherein only a limited amount of oxygen can be delivered, or gas generation would prove insufficient. Moreover, these approaches must address the second challenge, which pertains to ensuring size stability at the submicron level. This stability is critical for enhancing nanobubble accumulation in tumors, with size being the primary advantage of nanobubbles over their larger counterparts. Thirdly, achieving precise control over gas/drug release emerges as another critical aspect. This control is essential for tailoring responsive nanobubbles effectively to stimuli such as acoustic pressure, *a well-explored mechanism that has opened new avenues in both the diagnosis and therapy of cancer and other diseases*. Looking ahead, nanobubbles also present a range of other exogenously responsive mechanisms to environmental or external stimuli, such as light, temperature changes, electric and magnetic fields, and endogenous triggers like pH and redox states. These features align well with the versatile array of biomaterials (lipids, surfactants, polymers, and proteins), opening doors to diverse applications in theragnosis, drug design, and delivery. In summary, oxygenation by nanobubbles offers a promising strategy for addressing diseases characterized by hypoxia and mitochondrial dysfunction. It has the potential to improve oxygen delivery to specific tissues and enhance therapeutic outcomes while minimizing systemic side effects. However, addressing stability, biocompatibility, and regulatory concerns is critical for its successful translation into clinical practice. As research in this field advances, nanobubble-mediated oxygenation may emerge as a valuable tool in the treatment of various diseases.

## Figures and Tables

**Figure 1 nanomaterials-13-03060-f001:**
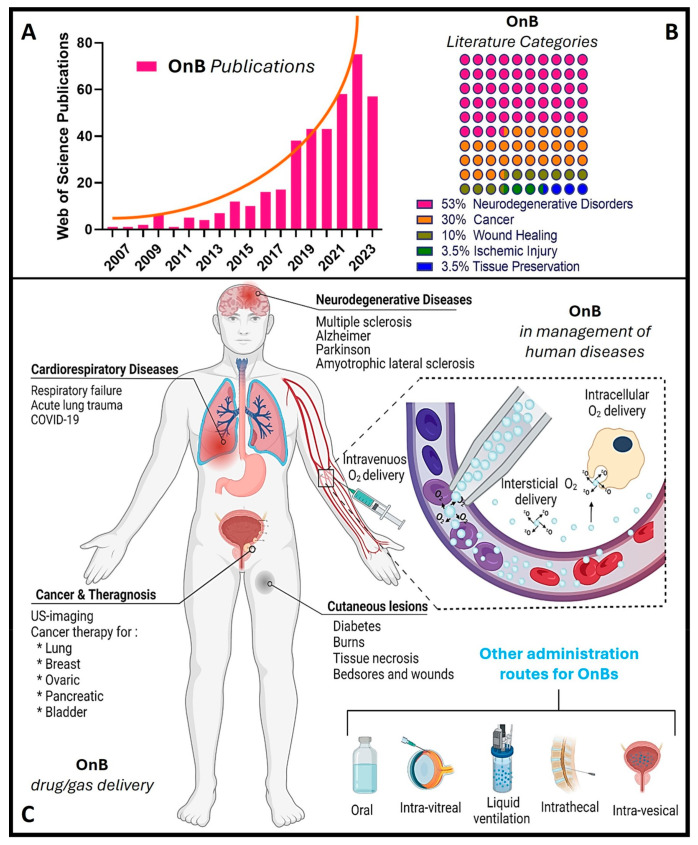
OnB (oxygen nanobubbles) and biomedicine research. (**A**) The number of articles published each year retrieved through a scientific search engine (Web of Science) using the term “oxygen nanobubble” followed by filtration/classification for application(s) in biomedicine. (**B**) The percentage of OnB articles published categorized by indication. (**C**) Illustration of drug/gas delivery based on OnB and their commonly investigated administration route(s) for the management of disease(s).

**Figure 2 nanomaterials-13-03060-f002:**
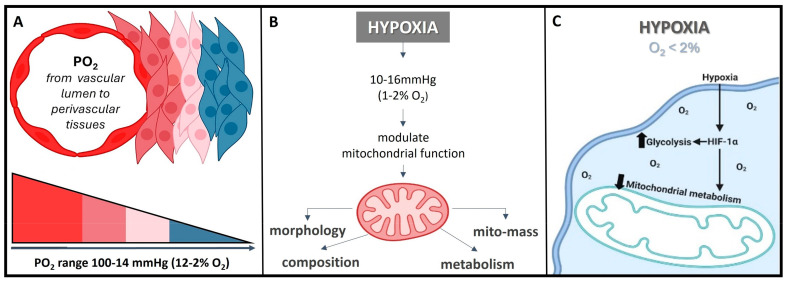
O_2_ need depends on cell niche amongst other demands. (**A**) Sites that are distant from major blood vessels are usually kept at a lower O_2_ level. However, low PO_2_ might still be higher than the PO_2_ of truly hypoxic conditions <16 mmHg (<2% O_2_). (**B**) Hypoxia acts on mitochondrial function, affecting the protein composition of the electron transport chain, mitochondrial morphology, and mitochondrial mass. (**C**) Hypoxia, besides playing a crucial role in influencing the cellular signaling pathways, gene expression, and the overall microenvironment, it regulates metabolic adaptation, promotes glycolysis, and decreases mitochondrial respiration, including HiF stabilization.

**Figure 3 nanomaterials-13-03060-f003:**
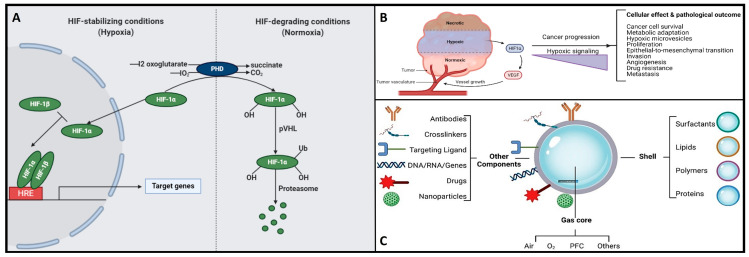
(**A**) The functional interplay of the HiF pathway in response to different oxygen concentrations. Herein, under hypoxic conditions, PHD is inhibited, and HiF-α is stabilized, which then translocates to the nucleus, dimerizes with its corresponding β-sub-unit, and recognizes the HREs motifs to induce the transcription of several target genes. On the other hand, when under normoxia, HiF1α-subunit prolyl residues are continuously hydroxylated by PHD. Following hydroxylation, pVHL binds and targets the HiF-1 α-subunit for ubiquitination and proteasomal degradation. (**B**) Hypoxic signaling and HiF-1α over-expression: a *key* hallmark in cancer progression. (**C**) Schematic representation of the structure and composition of engineering shelled micro/nanobubble systems.

**Figure 4 nanomaterials-13-03060-f004:**
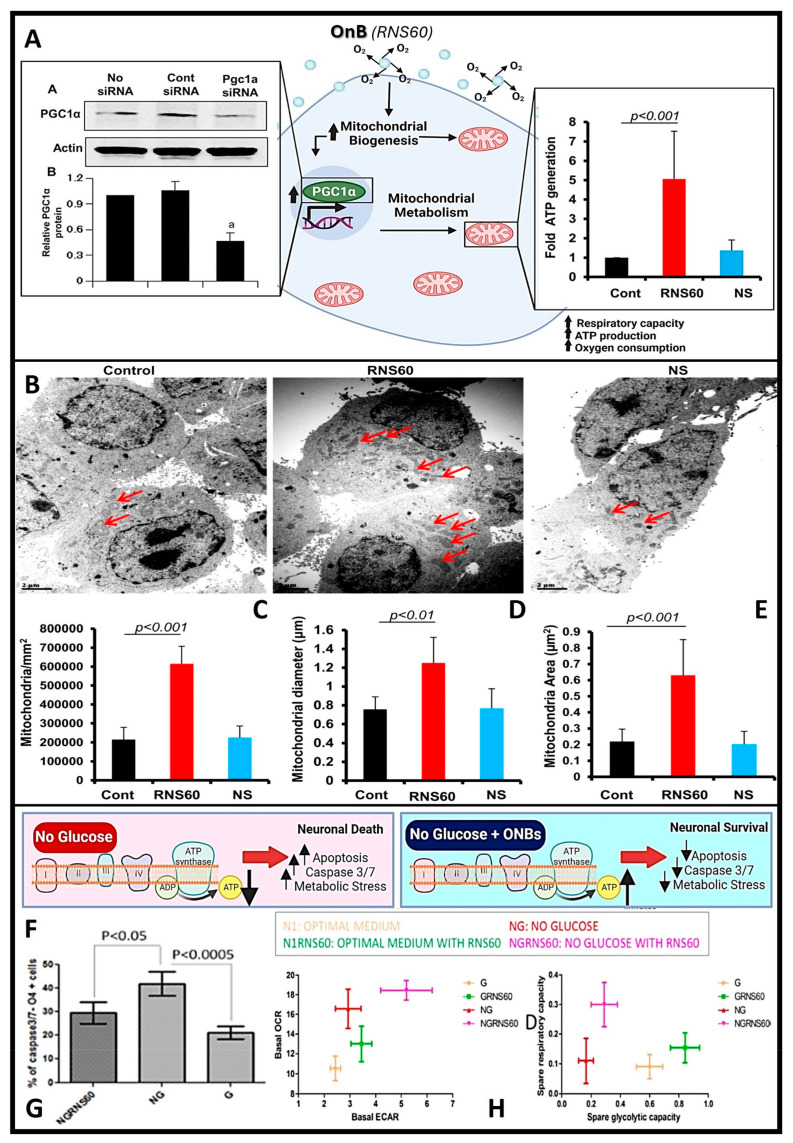
O_2_ nanobubble effect on mitochondrial bio-genesis, apoptosis, and metabolism. (**A**) Upregulation of PGC1α and ATP production in *RNS60*-treated MN9D mouse neuronal cells. Cells were transfected with control or PGC1α siRNAs. After 48 h of transfection, cells were treated with 10% v/v RNS60 for 4 h followed by monitoring the level of PGC1α by Western blot (a). Actin was run as loading control. Bands were scanned and values (PGC1α/Actin) presented as relative to control (b). Results (**B**) Cells treated with *RNS60* (10% *v*/*v*) and NS (10% *v*/*v*) under serum-free conditions for 24 h. Red lines indicate mitochondria and NS: normal saline. (**C**) *RNS60*, not NS, upregulated the number of mitochondria and (**D**) increased the mitochondrial diameter and (**E**) area in MN9D neuronal cells. (**F**) Glucose deprivation model in oligodendrocyte progenitor cells (OPCs) to assess OnB effect under metabolic stress. (**G**) Graph depicts the average % of total cells that were double positive for caspase3/7 and O4, and (**H**) Graph depicts the influence of *RNS60* on OCR vs. ECAR; Spare Respiratory Capacity vs. Spare Glycolytic Capacity in OLs cultured under optimal and glucose starvation conditions. *NG RNS60: No glucose with 10% v/v RNS60, NG: no glucose, G: with glucose, OCR: oxygen consumption rate, ECAR: extra-cellular acidification rate*. Reprinted (adapted) with permission from [5,21]. Copyright 2016 and 2017, Springer Nature.

**Figure 5 nanomaterials-13-03060-f005:**
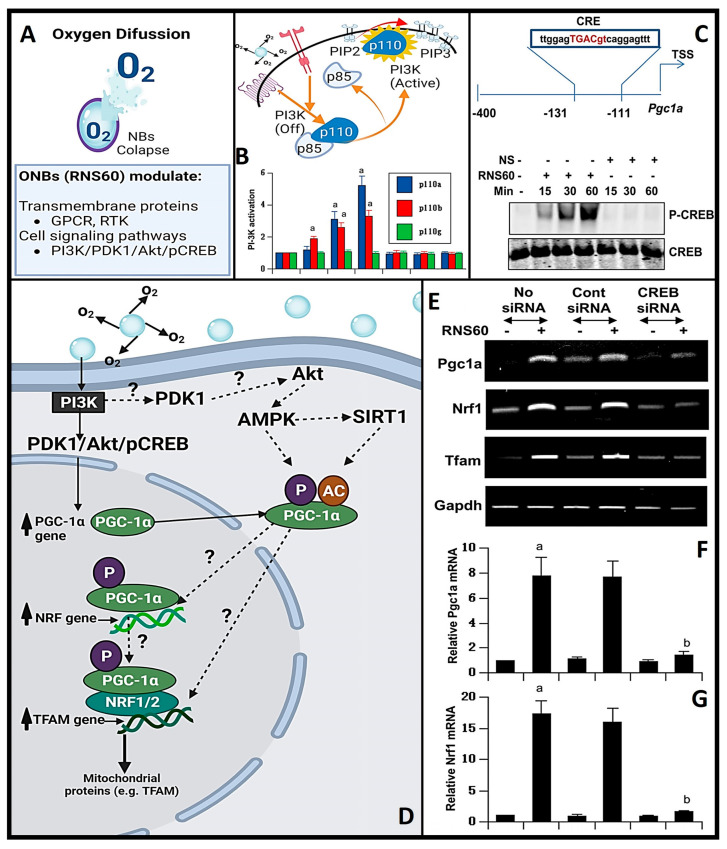
Upregulation of PGC1α and mitochondrial bio-genesis via PI3K/CREB in neuronal cells exposed to OnB. Cells were treated with 10% *v*/*v RNS60* or NS for different minute intervals or 2 h under serum-free conditions followed by monitoring the levels of PI3K (p110), phospho(P)-CREB, PGC1a, NRF1, and TFAM. (**A**) Illustrating OnB as modulators of transmembrane proteins and cellular signaling pathways. (**B**) Scheme of PI3K activation and levels of PI3K subunits p110α, p110β, and p110γ in cell membranes by Western blot analysis. Pan cadherin was run as a membrane marker. Data values (p110α/pan Cad, p110β/pan Cad and p110γ/pan Cad) are expressed relative to control: *p*-value < 0.001 vs. control. (**C**) Pgc1α gene promoter analysis shows the presence of a consensus cAMP response element (CRE) near the transcription start site (TSS). (**D**) Scheme of the upregulation of PGC1α, its upstream regulation (via PI3K/PDK1/AKT/CREB), their post-translation modifications (phosphorylation, acetylation) and its co-transcription via NRF1/2 to promote mitochondrial bio-genesis following an OnB treatment. The question mark (?) indicate limited evidence or further studies are required research (**E**) The mRNA expression of Pgc1α, NRF1, and TFAM by semi-quantitative RT-PCR and real-time PCR, (**F**) Pgc1a; (**G**) Nrf1. ^a^
*p*-value < 0.001 versus control; ^b^
*p*-value < 0.001 versus control siRNA-RNS60. Reprinted (adapted) with permission from [21]. Copyright 2017, Springer Nature.

**Figure 6 nanomaterials-13-03060-f006:**
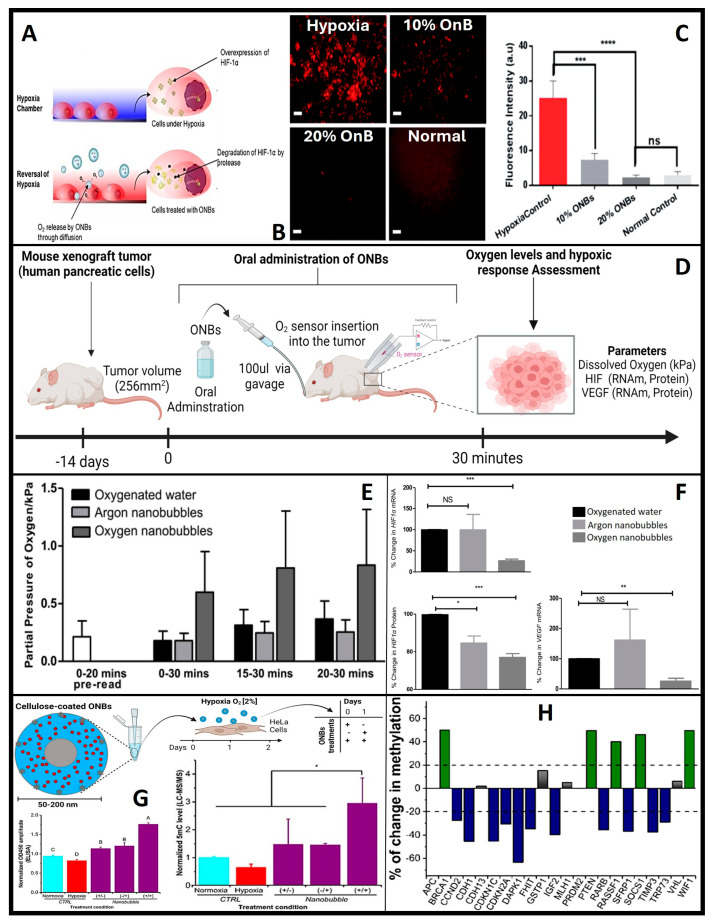
The in vitro and in vivo effect of OnB on modulating the hypoxic response(s) in/within the tumoral microenvironment (TME). (**A**) in vitro application of OnB for the reversal of hypoxia and degradation of HiF-1α expression. (**B**,**C**) Image-iT fluorescence of MDA-MB-231 cells under hypoxic conditions: *scale bar* = 20 µm. (**D**) Schematic representation of OnB and in vivo administration and its effects on oxygen pressure (**E**), and mRNA and protein levels of HiF-1Alpha and VEGF (**F**). * *p* < 0.05; ** *p* < 0.01; *** *p* < 0.001, **** *p* < 0.0001, NS = no significance. (**G**) Schematic representation illustrating a lipid-shelled or cellulose-coated OnB and its in vitro application for the reversal of hypoxia and degradation of HiF-1alpha alongside a comparison of HiF-1a expression and Image-iT fluorescence of MDA-MB-231 cells in control and after the reversal of hypoxic conditions. (**H**) OnB therapy triggers a significant reduction in promoter methylation for 11 of these 22 genes. Reprinted (adapted) with permission from [7,65,72]; Copyright 2018 (Taylor & Francis), Copyright 2016 (PLOS), and Copyright 2017 (Springer Nature), respectively.

**Table 1 nanomaterials-13-03060-t001:** OnB application in cancer studies.

Clinical Indication	Study Type	Oxygen Release Strategy	Delivery	Main Results and Conclusions
Lung cancer	in vitro	Diffusion	Cell Culture Media	Uncoated OnB reduce hypoxia-induced resistance in cancer cells [64].
Pancreatic cancer	in vitro/in vivo	UltraSound mediated	Oral	Surfactant-stabilized OnB reduce the transcriptional and protein levels of HIF1α [65]. Lipid-stabilized oxygen micro-bubbles and US reveal a 45% reduction in tumor volume five days after treatment [66].
Ovarian cancer	in vivo	UltraSound mediated	Intraperitoneal	Oxygen and Paclitaxel (PTX)-loaded lipid micro-bubbles downregulate HiF-1α and increase PTX effectivity [67].
Breast cancer	in vitro	Diffusion	Cell Culture Media	OnB revert hypoxia, downregulates HiF-1a, and improve cellular conditions, leading to further medical applications [7].
	in vivo	UltraSound mediated	Intraperitoneal	Oxygen micro-bubbles strongly enhance echo intensity in tumor and significantly enhance PO_2_ after US irradiation [53].
Chorio- carcinoma	in vitro	Diffusion	Cell Culture Media	Human JEG-3 cells showed a reduction (2-fold decrease—50%) in HiF-1α transcript levels at 3% O_2_ incubation [68].
NasoPharyngeal carcinoma	in vitro/in vivo	pH-responsive	Intravenous	pH-responsive OnB increase the intra-tumoral oxygen concentration six-fold, suggesting great potential for overcoming hypoxia-induced resistance [69].
Bladder/Colon Tumor Cell lines	in vitro/in vivo	UltraSound mediated	Intravesical Intravascular	OnB are a promising multi-modal and multifunctional strategy for imaging and targeting the hypoxic tumoral micro-environment [70,71]. The bladder tumoral PO_2_ increased by around 140% after the injection of OnB [72]. Biogenic nanobubbles (gas vesicles) enhance US contrast signal when compared to/with synthetic nanobubbles, enhancing tumor penetration with a range size of 2–200 nm [73].
Glioma	in vitro/in vivo	Photodynamic (PDT)	Intravenous	OnB, stimulated by function as an oxygen self-supplement agent, enhance the survival rate in the glioma-bearing mice model [33].

**Table 2 nanomaterials-13-03060-t002:** OnB application in neuroscience studies.

Clinical Indication	Study Type	Oxygen Release Strategy	Delivery	Main Results and Conclusions
Amyotrophic Lateral Sclerosis (ALS)	in vitro/in vivo	Diffusion	Intraperitoneal	*RNS60* protects neurons, decreasing ALS progression [37,74] and demonstrating the feasibility, safety, and tolerability of long-term administration of *RNS60* in patients with ALS [75].
	Clinical Trial	Diffusion	Intravenous	The effect of *RNS60* treatment on selected pharmacodynamic biomarkers in ALS patients was concurrently treated with riluzole (*NCT03456882*).
Multiple Sclerosis (MS)	in vitro/in vivo	Diffusion	Intraperitoneal and Nebulization	*RNS60* induced the activation of PI3K, promoting myelin gene transcription in oligodendrocytes (OL) and glial cells [76]. *RNS60* enhanced OL spare respiratory capacity (SRC) in response to metabolic stress (glucose-nutrient deprivation) [5]. *RNS60* led to the enrichment of anti-autoimmune regulatory T cells (Tregs) suppression of autoimmune Th17 cells [77].
Alzheimer’s disease (AD)	in vivo	Diffusion	Intraperitoneal	*RNS60* suppressed the hippocampus neuronal apoptosis and attenuated Tau phosphorylation and the burden of Ab [23]. *RNS60* upregulated the plasticity-related proteins (PSD95 and NR2A) and NMDA-dependent hippocampal calcium influx [21].
Parkinson’s disease (PD)	in vitro/in vivo	Diffusion	Intraperitoneal	*RNS60* enhance mitochondrial bio-genesis via PI3K/CREB and PGC1alpha in PD model [78]. Moreover, *RNS60* inhibited the activation of NF-κB in the SNpc of MPTP-intoxicated mice [78].
Spinal Cord Diseases, Injuries, and Compression	Clinical Trial	US Contrast	Intrathecal	The use of OnB and US improves the identification of discrete areas of perfusion changes in the spinal cord in subjects undergoing spinal cord decompression (*NCT05530798*).

**Table 3 nanomaterials-13-03060-t003:** OnB application in cardio–respiratory studies.

Clinical Indication	Study Type	Oxygen Release Strategy	Delivery	Main Results and Conclusions
Respiratory Failure and Acute Lung Trauma/Injury	in vitro/in vivo	Diffusion	Intraperitoneal Intravenous	Peritoneal microbubble oxygenation (PMO) provides extrapulmonary ventilation after complete tracheal occlusion [79]. PMO is also a promising strategy for other pulmonary diseases [6] as it oxygenates blood within 4 sec and does not cause hemolysis or complement activation in hypoxic rabbits [80].
Blood Oxygenation	in vitro/in vivo	Diffusion	Cell Culture Media	Oxygen microbubble-containing dextran solutions were effective for improving blood oxygenation [81].
	in vivo	Diffusion	Intravenous	Oxygen microbubbles are safe and effective in delivering more oxygen than human red blood cells (per gram) after being injected in vivo [82].
ExtraCorporeal Membrane Oxygenation (ECMO)	in vitro	Diffusion	De-oxygenated PBS	Protein-encapsulated oxygen microbubbles rapidly equilibrate hypoxia by releasing their oxygen core into an oxygen-depleted saline solution [83].
Anti-Thrombotic Effect	in vitro	US-mediated	Intravascular	PLA-combined Fe_3_O_4_-GO-ASA nanobubbles improve the anti-thrombin parameters and significantly inhibit thrombosis within rabbit blood [84].
MicroFluidic Device for Hypoxemia	in vitro/in vivo	Diffusion	Intravascular	OnB from the device was infused into the femoral vein, in vivo, wherein ∼20% of baseline VO_2_ can be delivered intravenously in real time [85].
Ischemic Stroke Re-Perfusion	in vivo	US-mediated	Intravascular	OnB with US stimulation provide sono-perfusion and local oxygen for the reduction in brain infarct size and neuroprotection after stroke re-perfusion [86].

**Table 4 nanomaterials-13-03060-t004:** OnB application in metabolic diseases, regenerative medicine, and molecular imaging studies.

Clinical Indication	Study Type	Oxygen Release Strategy	Delivery	Main Results and Conclusions
Diabetes, Burns, Tissue necrosis, Bedsores, and Wounds	in vitro/in vivo	US-mediated	Cell Culture Media	Dextran and chitosan nanobubbles might be proposed for the delivery of oxygen, which is enhanced by US with a frequency of 45 kHz in hypoxic-related diseases [2,87].
	Clinical Trial (*recruiting*)	Diffusion	Irrigation	This micro-/nano-bubble solution is suggested as an irrigation solution to improve wound oxygenation in ischemic tissues (*NCT05169814*).
Diabetic Retinopathy	in vitro/in vivo	Diffusion	Cell Culture Intravitreal	Dextran-OnB release 74.06 µg of O_2_ after 12 h at 37 °C and mitigate hypoxia during ischemic conditions in the eye upon timely administration [88].
Tissue Cutaneous Lesions	in vitro/in vivo	US-mediated	Cell Culture Media/Gel Formulation Topically Applied	Oxygen-loaded nano-droplets (OLNDs) are more effective than former oxygen-loaded nanobubbles enhancing oxy-hemoglobin levels by photoacoustic [89]. US-activated chitosan-shelled/DFP-cored OLNDs might be a novel, suitable, and cost-effective way to treat several hypoxia-associated pathologies of the cutaneous tissues [3].
Transdermal Drug Release	in vitro/in vivo	US-mediated	Transdermal Micro-needle Gel	The nanobubbles added into a micro-needle patch and used in addition to US cause better penetration and diffusion of drugs [90].
Cytocompatibility of OnB	in vitro	US-mediated	Cell Culture Media	Lipid-shelled/coated OnB show US imaging-responsiveness and enhance cell viability in several cell lines [15].
Drug Delivery in Fluids	in vitro	US-mediated	Hypoxic Solution	Oxygen release from polysaccharides–peptides, Pingxiao, and chitosan is 94.6%, 75.1%, and 40.2% (*respectively*) higher than water, especially under the US stimulus [91].
On-chip Contrast Agent for US Imaging	in vitro	US-mediated	Microfluidic	Micron-sized lipid shell-based perfluorocarbon gas microbubbles enhance the gas composition for US contrast agents with new shell materials [92].

## Data Availability

The datasets used and/or analyzed during the current study are available from the corresponding authors on request.

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
