# Peer review of "NanoBubble-Mediated Oxygenation: Elucidating the Underlying Molecular Mechanisms in Hypoxia and Mitochondrial-Related Pathologies"

_nanomaterials, 2023, doi:10.3390/nano13233060_

Round 1

Reviewer 1 Report

Comments and Suggestions for Authors

      In this research, the authors reviewed the recent development of nanobubble-mediated oxygenation: elucidating the underlying molecular mechanisms in hypoxia and mitochondrial-related pathologies. In my opinion, the current version of this manuscript fits the scope of Nanomaterials and could be accepted after minor revision.

My specific comments are in detail listed below:

1.     Some references are out of date (published before 2010). Some new references may be better. Besides, some minor mistakes exist in the references.

2.     In the introduction part, the authors should compare or point out the advantages or merits of nanobubble-mediated oxygenation. Some references should be added to this part including 10.1002/adma.202206121.

3.     Some figures are of low quality. A more clear version should be offered.

4.     In this review (Line 210-258), the authors should discuss merits of nanobubble-mediated oxygenation, especially hypoxia reversing mediated by mitochondrial respiration inhibition in tumors. Some references should be added to this part including 10.1002/advs.202207608.

5.     The usage of nanobubble-mediated oxygenation for disease treatments should be more clearly discussed.

6.     In conclusion, the clinical transformation barriers of nanobubble-mediated oxygenation for cancer treatment and disease treatment should be better out-looked.

Author Response

We extend our gratitude for granting us the chance to submit the revised manuscript (nanomaterials-2687317) titled "NanoBubble-mediated Oxygenation: Elucidating the Underlying Molecular Mechanisms in Hypoxia and Mitochondrial-related Pathologies" to Nanomaterials.

Your commitment of time and effort in offering constructive feedback on our manuscript is highly appreciated. We value the insightful comments provided, and we have diligently incorporated the recommended changes to enhance the overall quality of the paper. Additionally, we have denoted most of the revisions within the manuscript by highlighting them in red.

Below is a point-by-point response to your comments:

Comments from Reviewer 1

General comments

“In this research, the authors reviewed the recent development of nanobubble-mediated oxygenation: elucidating the underlying molecular mechanisms in hypoxia and mitochondrial-related pathologies. In my opinion, the current version of this manuscript fits the scope of Nanomaterials and could be accepted after minor revision”.

Comment 01:

Some references are out of date (published before 2010). Some new references may be better. Besides, some minor mistakes exist in the references.

Response:

Thank you for pointing this out. We agree with this comment. We have reinforced the manuscript adding new/updated references that reflect a more current state of the review. We move from 105 citation to 120 in total, incorporated into tables as well as in main text. Therefore, we kept some of the early studies due to value of laying the foundation to build a complete overview from earliest discovery, development to recent approaches of/for using nanobubbles for oxygenation purposes. To rephrase, considering that nanobubbles are relatively known as an emerging technology, the inclusion of some studies before 2010 can be deemed beneficial for the review, especially with the compilation and discussion of many studies used for hypoxic or mitochondrial related disease, which were mainly analyzed and summarized in the prepared tables, hence, from past to present.

Comment 02:

“In the introduction part, the authors should compare or point out the advantages or merits of nanobubble-mediated oxygenation. Some references should be added to this part including 10.1002/adma.202206121.”.

Response:

We express our deepest appreciation for this insightful comment, as it significantly strengthens our perspective on the importance of highlighting the benefits of nanobubble platforms in selectively promoting metabolic re-programming within impaired cell niches or cellular microenvironments associated with diseases such as cancer, neurodegenerative disorders, and chronic wound healing. In response to this comment, we have incorporated an additional paragraph to the introduction, as suggested/requested,  that underscores the significance of developing selective metabolic reprogramming, particularly in the context of targeting cancer cells. This involves disrupting their ability to adapt to low oxygen conditions, inhibiting specific metabolic pathways (e.g., glycolysis or glutaminolysis), and modulating the activity of key enzymes or oxygen sensors to regulate the hypoxia-inducible factor (HiF) system. Such interventions can effectively reduce the survival and proliferation of cancer cells, rendering them more susceptible to traditional therapeutic modalities such as chemotherapy and radiotherapy. Kindly refer to the paragraph in red between lines (98-108), for an example of the incorporated comparison and pointing out of the advantages and merits of nanobubble-mediated oxygenation, in the this context.

Comment 03:

“Some figures are of low quality. A more clear version should be offered.

Response:

We thank you for the suggestions. We went ahead and revised all figures and did our best of abilities to improve the quality and resolution of images. Figure 1 was modified with an updated graph of OnB-related publication history/evolution.

Comment 04:

“ In this review (Line 210-258), the authors should discuss merits of nanobubble-mediated oxygenation, especially hypoxia reversing mediated by mitochondrial respiration inhibition in tumors. Some references should be added to this part including 10.1002/advs.202207608”.

Response:

Agreed. We have, accordingly, carried out this suggestion. In section 2 of the manuscript, we added a detailed discussion/other remarks, in red, addressing how hypoxia reversing is also mediated by mitochondrial respiration inhibition in tumors. Beside we included the suggested paper(s) in the references list, alongside others.

Comment 05:

“The usage of nanobubble-mediated oxygenation for disease treatments should be more clearly discussed”.

Response:

As suggested, we added a discussion/other remarks section addressing in detail the reviewer’s comment(s). Beside of the previously discussed section in this review of how nanobubble-mediated oxygenation works in terms of intracellular signalingn and subcellular (mitochondrial level),  we also discussed its relevance in/to specific diseases and/or medical conditions, where nanobubble-mediated oxygenation has shown/exhibited promise (describing how this technology has been applied in the treatment of conditions like cancer, chronic wounds, or neurodegenerative diseases). Moreover, we mentioned the current challenges, limitations and  perspectives. Finally, we also modified a statement, clarifying that perfluorocarbon is not a gas; described as follows: Liquid perfluorocarbons (PFC), sulfur hexafluoride (SF6), C3F8, and NO mixed with oxygen also have been used to formulate microbubbles [29,62,63].

Comment 06:

“  In conclusion, the clinical transformation barriers of nanobubble-mediated oxygenation for cancer treatment and disease treatment should be better out-looked”

Response:

We improved our conclusions considering the clinical transformation barriers of nanobubbles-mediated oxygenation, as suggested.

 The authors thank you Reviewer 1 for your invaluable insights to further improve our work.

Reviewer 2 Report

Comments and Suggestions for Authors

The manuscript is well written and organized contents are also good. The submitted manuscript can be accepted for publication in Nanomaterials. However, before publication, the following comments should be incorporated in the revised manuscript.

1.      Several reports based on the nanobubbles are published in 2023, however authors only provided few 2023 studies. Therefore, strong recommendation to emphasize the importance of latest published articles in 2023.

2.      In the introduction section, consider presenting a Year-wise Published Paper Distribution graph related to NanoBubble-mediated Oxygenation, covering the past 10 years of research articles as indicated in (https://doi.org/10.1021/acs.chemrev.8b00539), Figure 1.

Comments on the Quality of English Language

Moderate English editing required.

Author Response

We extend our gratitude for granting us the chance to submit the revised manuscript (nanomaterials-2687317) titled "NanoBubble-mediated Oxygenation: Elucidating the Underlying Molecular Mechanisms in Hypoxia and Mitochondrial-related Pathologies" to Nanomaterials.

Your commitment of time and effort in offering constructive feedback on our manuscript is highly appreciated. We value the insightful comments provided, and we have diligently incorporated the recommended changes to enhance the overall quality of the paper. Additionally, we have denoted most of the revisions within the manuscript by highlighting them in red.

Below is a point-by-point response to your comments:

Comments from Reviewer 2

General comments

“The manuscript is well written and organized contents are also good. The submitted manuscript can be accepted for publication in Nanomaterials. However, before publication, the following comments should be incorporated in the revised manuscript”.

Comment 01:

“Several reports based on the nanobubbles are published in 2023, however authors only provided few 2023 studies. Therefore, strong recommendation to emphasize the importance of latest published articles in 2023.In this research, the authors reviewed the recent development of nanobubble-mediated oxygenation: elucidating the underlying molecular mechanisms in hypoxia and mitochondrial-related pathologies. In my opinion, the current version of this manuscript fits the scope of Nanomaterials and could be accepted after minor revision”

Response:

We added 15 recent papers improving the current state of the topic addressed in this work. Thereby, we move from 105 citations to 120 in total, taking the reviewer’s comment and suggestion into consideration.

Comment 02:

In the introduction section, consider presenting a Year-wise Published Paper Distribution graph related to NanoBubble-mediated Oxygenation, covering the past 10 years of research articles as indicated in (https://doi.org/10.1021/acs.chemrev.8b00539), Figure 1.

Response:

We edit the format of the publications by year, thereby extending to since 2006 until 2023. Hence, a year-wise published paper distribution graph is now included in Figure 1, as the reviewer suggested.